# Statistical Mechanics of Discrete Multicomponent Fragmentation

**Themis Matsoukas** 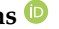

Department of Chemical Engineering, Pennsylvania State University, University Park, PA 16802, USA; txm11@psu.edu; Tel.: +1-814-863-2002

**Abstract:** We formulate the statistics of the discrete multicomponent fragmentation event using a methodology borrowed from statistical mechanics. We generate the ensemble of all feasible distributions that can be formed when a single integer multicomponent mass is broken into fixed number of fragments and calculate the combinatorial multiplicity of all distributions in the set. We define random fragmentation by the condition that the probability of distribution be proportional to its multiplicity, and obtain the partition function and the mean distribution in closed form. We then introduce a functional that biases the probability of distribution to produce in a systematic manner fragment distributions that deviate to any arbitrary degree from the random case. We corroborate the results of the theory by Monte Carlo simulation, and demonstrate examples in which components in sieve cuts of the fragment distribution undergo preferential mixing or segregation relative to the parent particle.

**Keywords:** discrete fragmentation; multicomponent; partition function; multiplicity of distribution

## 1. Introduction

Objects disintegrate into fragments via impact, detonation, degradation, or cleavage of the bonds that hold the structure together. The object in question may range from the sub-atomic [1,2] and molecular [3–5] to living organisms [6], social structures [7,8], and celestial bodies [9], a diversity of scale and physics that is united by a common mathematical formalism. At its core, fragmentation is a branching process in which a parent object ("particle") produces a set of offspring. The evolution of a population that undergoes splitting of this form is given by the fragmentation equation, an integro-differential equation that accounts for the generation and depletion of sizes due to fragmentation [10,11]. The primary input to this formulation is a breakup model that specifies the distribution of fragments produced by a given parent size and the relative rate at which different sizes break up. This population balance approach forms the basis for the mathematical treatment and numerical modeling of fragmentation in granular, colloid, and polymeric systems [11–18]. The mathematical literature of the fragmentation equation is rich and focuses on analytic solutions, existence criteria, and stability. Of particular interest is the emergence of "shattering", a process akin to a phase transition that is demonstrated through the appearance of a finite population of particles with zero mass [19–22]. An alternative approach views fragmentation as the disintegration of bonds between the constitutive units of the particle and uses percolation theory to model and simulate the breakup of systems with topological structure. In contrast to the population balance method, which is a mean-field method, percolation treats fragmentation at the discrete probabilistic level [7,8,23]. Other treatments view fragmentation in a more abstract way as a partitioning of a discrete event space and use combinatorial and probabilistic methods to obtain the partition function and the mean distribution in this space [24–27].

A central question in fragmentation is the distribution of fragments per fragmentation event. The most common theoretical model is that of random binary fragmentation. In this model a parent cluster produces two fragments with uniform probability [22]. Empirical models have been proposed for the breakage of a single component into multiple pieces of unequal size and typically require a set of parameters that control the shape of the fragment distribution [28,29]. Systems of practical interest are almost always multicomponent. Pharmaceutical granulation is a case in point: granules contain an active pharmaceutical ingredient, an inert excipient, binder, and are characterized by additional attributes such as porosity or shape factor that behave as pseudo components [30]. Nonetheless, no generalized approaches exist to treat the problem of multicomponent fragmentation into arbitrary number of fragments. The mathematical treatment of multicomponent fragmentation into any number of pieces cannot be simply obtained as an extension of the one-component problem. In addition to the size distribution of fragments one must consider the distribution of components, provide rules for apportioning components to the fragments, offer a definition of what is meant by "random fragmentation" when both size and composition are distributed, and provide the means for constructing models that deviate from the random case to any extent desired.

The purpose of this paper is to address these questions by formulating the statistics of a single fragmentation event in the discrete domain for arbitrary number of fragments and components in a way that is general and not bound by the details of the particular application. Our interest is not in the physics behind the splitting of an object into smaller parts, but rather in the probabilistic treatment of the partitioning itself under the constraint of a conservation law, in this case conservation of mass. The main idea is this. We start with a multicomponent particle that is made of discrete units of any number of components, subject it to one fragmentation event with fixed number of fragments, and construct the set of all fragment distributions that can be obtained in this manner. We calculate the partition function of this ensemble of fragments, assign probabilities in proportion to the multiplicity of each distribution, and obtain the mean distribution in terms of the partition function. We then introduce a bias functional that biases the distribution away from that of random fragmentation. We present results from Monte Carlo simulations to corroborate the theory and show that components may preferentially mix or unmix in the fragments depending on the choice of the bias functional.

## 2. Random Fragmentation

### 2.1. One-Component Random Fragmentation

In discrete fragmentation, a particle composed of $M$ integer units breaks up into $N$ fragments, $\{m_1, m_2 \cdots m_N\}$ that satisfy the mass balance condition

$$\sum_{i=1}^{N} m_i = M. \tag{1}$$

The distribution of fragments is given by vector $\mathbf{n} = (n_1, n_2 \cdots)$ whose element $n_i$ is the number of fragments that contain $i$ units of mass. We suppose that $N$ is fixed but $\mathbf{n}$ is not; that is, if the fragmentation event is repeated with an identical parent particle the distribution of fragments may be different but the total number of fragments is always $N$. We refer to this process as $N$-nary fragmentation. All fragment distributions produced by this mechanism satisfy the following two conditions:

$$\sum_{i=1}^{\infty} n_i = N, \tag{2}$$

$$\sum_{i=1}^{\infty} i n_i = M. \tag{3}$$

The first condition states that the number of fragments is $N$; the second that their mass is equal to the mass of the parent particle. Conversely, any distribution that satisfies the above two equations is a feasible distribution of fragments by $N$-nary fragmentation of mass $M$. Thus the set $\mathscr{E}_{M;N}$ of all distributions that satisfy Equation (2) forms the ensemble of fragment distributions produced from $M$. We assign a probability $P(\mathbf{n})$ on the distributions in $\mathscr{E}_{M;N}$, normalized over all distributions that satisfy Equations (2) and (3). Our goal is to obtain this distribution under various fragmentation models.

We will call the process random fragmentation if all ordered lists of $N$ fragments produced by the same mass are equally probable. This views the ordered list of fragments, which we call configuration, as the elementary stochastic variable in this problem.

### 2.1.1. Probability of Random Fragment Distribution

**Proposition 1.** *The probability of distribution* $\mathbf{n}$ *produced by random N-nary fragmentation of mass M is*

$$P(\mathbf{n}) = \frac{\mathbf{n}!}{\binom{M-1}{N-1}}, \tag{4}$$

*where* $\mathbf{n}! = (n_1, n_2 \cdots)!$ *is the multinomial coefficient of distribution* $\mathbf{n}$,

$$\mathbf{n}! = \frac{(\sum_i n_i)!}{\prod_i n_i!} = \frac{N!}{n_1! n_2! \cdots}. \tag{5}$$

**Proof.** First we note that the number of ordered lists that can be formed by breaking integer $M$ into $N$ fragments is

$$\Omega_{M;N}^{(1)} = \binom{M-1}{N-1}. \tag{6}$$

This is the number of ways to partition integer $M$ into $N$ parts and can be shown easily as follows [31]: thread $M$ balls into a string and partition them into $N$ pieces by cutting the string at $N-1$ points (Figure 1). There are $M-1$ points where we can cut and must choose $N-1$ of them. The number of ways to do this is the binomial factor on the RHS of Equation (6).

If all ordered lists of fragments are equally probable, the probability of ordered list $\mathbf{m} = (m_1, m_2 \cdots m_N)$ is

$$\text{Prob}(\mathbf{m}) = \frac{1}{\Omega_{M;N}^{(1)}}. \tag{7}$$

There are $\mathbf{n}!$ ordered lists with the same distribution of fragments $\mathbf{n}$. Accordingly, the probability of $\mathbf{n}$ is

$$P(\mathbf{n}) = \mathbf{n}! \, \text{Prob}(\mathbf{m}) = \frac{\mathbf{n}!}{\Omega_{M;N}^{(1)}}. \tag{8}$$

This proves the proposition. □

The multinomial factor $\mathbf{n}!$ is the multiplicity of distribution $\mathbf{n}$, namely, the number of configurations (ordered lists of fragments) represented by $\mathbf{n}$. Using $\omega(\mathbf{n}) = \mathbf{n}!$ to notate this multiplicity, the probability of distribution is expressed as

$$P(\mathbf{n}) = \frac{\omega(\mathbf{n})}{\Omega_{M;N}^{(1)}}, \tag{9}$$

and $\Omega^{(1)}$ satisfies

$$\sum_{\mathbf{n}} \omega(\mathbf{n}) = \Omega_{M;N}^{(1)}. \tag{10}$$

The summation is over all distributions $\mathbf{n} \in \mathscr{E}_{M;N}$, namely, over all distributions that satisfy Equations (2) and (3). Accordingly, $\Omega_{M;N}^{(1)}$ is the total multiplicity in the ensemble, equal to the number of ordered configurations of fragments that can be produced from integer mass $M$ breaking into $N$ fragments. We refer to $\Omega_{M;N}^{(1)}$ as the partition function of the one-component ensemble of fragments.

**Figure 1.** Random fragmentation of integer mass $M$ into $N$ pieces is equivalent to breaking a string with $M$ beads at $N-1$ random points. With $M = 10$, $N = 3$ the number of possible partitions is $\binom{9}{2} = 36$. If the mass is made up of two colors, every permutation of the beads is equally probable; with $M_A = 6$, $M_B = 4$ the number of partitions increases by the factor $\binom{6+4}{4} = 210$ and the total number of permutations is 7560.

2.1.2. Mean Fragment Distribution

Each distribution $\mathbf{n}$ appears in the ensemble of fragment distributions with probability $P(\mathbf{n})$; the mean distribution of fragments is their ensemble average:

$$\langle \mathbf{n} \rangle = \sum_{\mathbf{n}} \mathbf{n} \, P(\mathbf{n}), \tag{11}$$

with $P(\mathbf{n})$ from Equation (4) and with the summation going over all distributions that are produced by $N$-nary fragmentation of integer mass $M$.

**Proposition 2.** *The mean distribution in N-nary random fragmentation is*

$$\frac{\langle n_k \rangle}{N} = \frac{\Omega_{M-k;N-1}^{(1)}}{\Omega_{M;N}^{(1)}} = \binom{M-k-1}{N-2} \Big/ \binom{M-1}{N-1}, \tag{12}$$

*with $M \geq N \geq 2$ and $k = 1, \cdots M - N + 1$.*

**Proof.** First we write the probability of distribution in the form

$$P(\mathbf{n}) = \frac{N!}{\Omega_{M;N}^{(1)}} \prod_{i=1}^{\infty} \frac{\alpha_i^{n_i}}{n_i!} \tag{13}$$

with $\alpha_i > 0$ and note that this reverts to Equation (4) when $\alpha_i = 1$. We will retain the factors $\alpha_i$ and will set them equal to 1 at the end. The normalization condition on the probability $P(\mathbf{n})$ reads

$$\Omega_{M;N}^{(1)} = N! \sum_{\mathbf{n}} \prod_{i=1}^{\infty} \frac{\alpha_i^{n_i}}{n_i!}. \tag{14}$$

The derivative of $\log \Omega_{M;N}^{(1)}$ with respect to $\alpha_k$ is

$$\frac{\partial \log \Omega_{M;N}^{(1)}}{\partial \alpha_k} = \frac{N!}{\alpha_k \Omega_{M;N}^{(1)}} \sum_{\mathbf{n}} n_k \prod_i \left( \frac{\alpha_i^{n_i}}{n_i!} \right) = \frac{\langle n_k \rangle}{\alpha_k}, \tag{15}$$

where $\langle n_k \rangle$ is the mean value of $n_k$ in the ensemble of fragments. We also have

$$\frac{\partial \Omega^{(1)}_{M;N}}{\partial \alpha_k} = N \left\{ (N-1)! \sum_{\substack{\mathbf{n} \\ n_k \neq 0}} \left( \cdots \frac{\alpha_i^{n_k-1}}{(n_k-1)!} \cdots \right) \right\} = N \, \Omega^{(1)}_{M-k;N-1}. \tag{16}$$

The summand in the expression in the middle amounts to removing one fragment of mass $k$ from all distributions of the ensemble; accordingly, the quantity in braces is the partition function $\Omega^{(1)}_{M-k;N-1}$. Combining Equations (15) and (16) and setting $\alpha_k = 1$ we obtain Equation (12). A similar proof was given by Durrett et al. [25] for a closely related system. $\square$

Equation (12) was previously obtained by Montroll and Simha [12] via a combinatorial derivation. Notably, it is the same distribution as in discrete binary aggregation (the reverse process of binary fragmentation) with constant kernel, derived by Hendriks et al. [24] who also credit older unpublished work by White. It also appears outside the context of fragmentation when the probability distribution is of the form of Equation (13) (see for example [32]).

For large $M$ the fragment distribution becomes

$$\langle n_k \rangle \rightarrow \frac{N(N-1)}{M} \left( 1 - \frac{k}{M} \right)^{N-2}. \tag{17}$$

This is the continuous limit of random fragmentation of a straight line into $N$ segments, an elementary result that has been derived multiple times in the literature. The earliest report known to us is by Feller [33] who corrected an earlier approximation by Ruark [34].

*2.2. Two-Component Random Fragmentation*

2.2.1. Representations of Bicomponent Populations

We now consider a particle that is made of two components. The particle contains $M_A$ units of component $A$, $M_B$ units of component $B$, and its mass is $M = M_A + M_B$. The distribution of fragments is given by the two-dimensional vector $\mathbf{n} = \{n_{a,b}\}$ where $n_{a,b}$ is the number of fragments that contain $a$ units of $A$ and $b$ units of $B$. This distribution satisfies the conditions

$$\sum_{a=0}^{\infty} \sum_{b=0}^{\infty} n_{a,b} = N, \tag{18}$$

$$\sum_{a=0}^{\infty} \sum_{b=0}^{\infty} a \, n_{a,b} = M_A, \tag{19}$$

$$\sum_{a=0}^{\infty} \sum_{b=0}^{\infty} b \, n_{a,b} = M_B. \tag{20}$$

The set $\mathscr{E}_{M_A,M_B;N}$ of all distributions that satisfy the above conditions constitutes the set of feasible distributions in bicomponent fragmentation. Strictly speaking, the upper limit in these summations is constrained by $a \leq M_A, b \leq M_B$. Under the convention that $n_{a,b} = 0$ outside the meaningful range of $a$ and $b$, we may set the upper limit to $\infty$.

The color-blind size distribution or simply "size distribution" $\mathbf{n}_{A+B} = \{n_k\}$ is the distribution of the mass of the fragments $k = a + b$ regardless of composition:

$$n_k = \sum_{a=0}^{k} n_{a,k-a}, \quad k = 1, 2 \cdots \tag{21}$$

and satisfies the conditions

$$\sum_{k=1}^{\infty} n_k = N, \tag{22}$$

$$\sum_{k=1}^{\infty} k n_k = M_A + M_B = M. \tag{23}$$

These are the same as Equations (2) and (3) in the one-component case for a particle with mass $M_A + M_B$. Accordingly, the feasible set of the color-blind distribution is $\mathscr{E}_{M;N}$ with $M = M_A + M_B$.

The sieve-cut distribution $\mathbf{n}_{A|k} = \{n_{a|k}\}$ is the number of fragments with size $k$ that contains $a$ units of component $A$:

$$n_{a|k} = n_{a,k-a}, \quad (a = 1 \cdots k, \ k = 1 \cdots M). \tag{24}$$

and satisfies the normalizations

$$\sum_{k=1}^{\infty} \sum_{a=0}^{k} n_{a|k} = N,$$

$$\sum_{k=1}^{\infty} \sum_{a=0}^{k} a n_{a|k} = M_A,$$

$$\sum_{k=1}^{\infty} \sum_{a=0}^{k} k n_{a|k} = M_A + M_B.$$

We divide the sieve-cut distribution by the number of fragments of size $k$ to obtain the compositional distribution of component $A$ within fragments of fixed size $k$,

$$c_{a|k} = \frac{n_{a|k}}{n_k}. \tag{25}$$

The compositional distribution is normalized to unity and may be interpreted as the conditional probability to obtain a fragment with $a$ units of $A$, given that the fragment has mass $k$. The bicomponent distribution may now be expressed in terms of the color-blind distribution $\mathbf{n}_{A+B}$ and the compositional distribution $c_{a|k}$ in the form

$$n_{a,k-a} = n_k \, c_{a|k}. \tag{26}$$

If we divide both sides by the total number of fragments, the result reads as a joint probability: the probability $n_{a,k-a}/N$ to obtain a fragment with mass $k$ that contains $a$ units of component $A$ is equal to the probability $n_k/N$ to obtain a fragment of mass $k$ times the probability $c_{a|k}$ to obtain a fragment with $a$ units of component $A$ given that the mass of the fragment is $k$.

### 2.2.2. The Ensemble of Random Fragment Distributions

Random fragmentation is implemented by analogy to the one-component case: we line up the unit masses in the particle into a string and cut at $N - 1$ places. Every cut is equally probable and so is every permutation in the order of the beads.

**Proposition 3.** *The probability of fragment distribution* $\mathbf{n}$ *in random bicomponent fragmentation is*

$$P(\mathbf{n}) = \frac{\mathbf{n}!}{\Omega_{M_A,M_B;N}^{(2)}} \prod_{a=0}^{\infty} \prod_{b=0}^{\infty} \binom{a+b}{a}^{n_{a,b}}, \tag{27}$$

*where* $\mathbf{n}!$ *is the multinomial coefficient of the bicomponent distribution,*

$$\mathbf{n}! = \frac{N!}{\prod_{a=0}^{\infty} \prod_{b=0}^{\infty} n_{a,b}!} \tag{28}$$

and $\Omega^{(2)}_{M_A,M_B;N}$ is the two-component partition function, given by

$$\Omega^{(2)}_{M_A,M_B;N} = \binom{M_A + M_B}{M_A} \Omega^{(1)}_{M_A+M_B;N}. \tag{29}$$

**Proof.** First we count the number of ordered sequences of fragments (configurations). Configurations are distinguished by the order the fragments and by the order of components within fragments (Figure 1). We color the components and place them in a line in some order. There are $M_A$ $A$s and $M_B$ $B$s; the number of permutations is $\binom{M_A+M_B}{M_A}$. Each permutation produces $\Omega^{(1)}_{M_A+M_B;N}$ configurations with $\Omega^{(1)}$ given in Equation (6). The total number of configurations, therefore, is their product and proves Equation (29).

Since all configurations are equally probable, the probability of fragment distribution **n** is proportional to the number of configurations with that distribution. This is equal to the number of permutations in the order of the fragments times the number of permutations in the order of components within the fragments. The number of permutations in the order of fragments is given by the multinomial factor of bicomponent distribution in Equation (28). The number of permutations of components within a fragment that contains $a$ units of $A$ and $b$ units of $B$ is $\binom{a+b}{a}$ and since there are $n_{a,b}$ such fragments, the total number of internal permutations in distribution **n** is

$$\prod_{a=0}^{\infty}\prod_{b=0}^{\infty} \binom{a + b}{a}^{n_{a,b}}. \tag{30}$$

The probability of distribution **n** is equal to the product of Equations (28) and (30) divided by the total number of configurations, given by Equation (29). The result is Equation (27) and proves the proposition. □

As a corollary, we obtain the multiplicity of the bicomponent distribution,

$$\omega(\mathbf{n}) = \mathbf{n}! \prod_{a=0}^{\infty}\prod_{b=0}^{\infty} \binom{a + b}{a}^{n_{a,b}}. \tag{31}$$

Thus, we write

$$P(\mathbf{n}) = \frac{\omega(\mathbf{n})}{\Omega^{(2)}_{M_A,M_B;N}} \tag{32}$$

with $\Omega^{(2)} = \sum_{\mathbf{n}} \omega(\mathbf{n})$.

An alternative result for $P(\mathbf{n})$ is obtained by expressing the bicomponent distribution **n** in terms of the color-blind distribution $\mathbf{n}_{A+B}$ and all sieve-cut distributions $\mathbf{n}_{A|k}$. The result is

$$P(\mathbf{n}) = \frac{\mathbf{n}_{A+B}!}{\Omega^{(2)}_{M_A,M_B;N}} \prod_{k=1}^{\infty} \left\{ \mathbf{n}_{A|k}! \prod_{a=0}^{k} \binom{k}{a}^{n_{a|k}} \right\} \tag{33}$$

and is based on the identity

$$\mathbf{n}! \prod_{a=0}^{\infty}\prod_{b=0}^{\infty} \binom{k}{a}^{n_{a,b}} = \mathbf{n}_{A+B}! \prod_{k=0}^{\infty} \left\{ \mathbf{n}_{A|k}! \prod_{a=0}^{k} \binom{k}{a}^{n_{a|k}} \right\}. \tag{34}$$

Here, $\mathbf{n}_{A+B}!$ is the multinomial coefficient of the color-blind distribution,

$$\mathbf{n}_{A+B}! = \frac{N!}{n_1! \, n_2! \cdots} \tag{35}$$

and $\mathbf{n}_{A|k}!$ is the multinomial coefficient of the sieve-cut distribution,

$$\mathbf{n}_{A|k}! = \frac{n_k!}{n_{0|k}! \, n_{1|k}! \cdots n_{k|k}!}. \tag{36}$$

### 2.2.3. Mean Fragment Distribution

**Proposition 4.** *The mean distribution of fragments in random bicomponent fragmentation is*

$$\frac{\langle n_{a,b} \rangle}{N} = \binom{a+b}{a} \frac{\Omega^{(2)}_{M_A-a,M_B-b;N-1}}{\Omega^{(2)}_{M_A,M_B;N}} \tag{37}$$

**Proof.** The proof follows in the steps of the one-component problem. We express the multiplicity and the partition function in the form

$$\omega(\mathbf{n}) = N! \prod_{a=0}^{\infty} \prod_{a=b}^{\infty} \frac{\alpha_{a,b}^{n_{a,b}}}{n_{a,b}!}, \tag{38}$$

$$\Omega^{(2)}_{M_A,M_B;N} = N! \sum_{\mathbf{n}} \prod_{a=0}^{\infty} \prod_{a=b}^{\infty} \frac{\alpha_{a,b}^{n_{a,b}}}{n_{a,b}!} \tag{39}$$

With $\alpha_{a,b} = \binom{a+b}{a}$ we recover the result for random fragmentation but for the derivation we treat $\alpha_{a,b}$ as a variable. Following the same procedure that led to Equation (12) we now obtain

$$\frac{\langle n_{a,b} \rangle}{N} = \alpha_{a,b} \frac{\Omega^{(2)}_{M_A-a,M_B-b;N-1}}{\Omega^{(2)}_{M_A,M_B;N}}. \tag{40}$$

To arrive at this result, we note that differentiation of the partition function with respect to $\alpha_{a,b}$ by analogy to Equation (16) amounts to removing one cluster that contains $a$ units of $A$ and $b$ units of $B$, thus producing the partition function $\Omega^{(2)}_{M_A-a,M_B-b;N-1}$ in the numerator of Equation (40). Setting $\alpha_{a,b} = \binom{a+b}{a}$ we obtain Equation (37). □

**Alternative Proof.** An alternative proof will be obtained by a mean-field argument. First we write the mean distribution in the form

$$\langle n_{a|k} \rangle = \langle n_k \rangle \overline{c_{a|k}}, \tag{41}$$

where $\langle n_{a|k} \rangle$ and $\langle n_k \rangle$ are the ensemble averages of $n_{a|k}$ and $n_k$, respectively, and $\overline{c_{a|k}} = \langle n_{a|k} \rangle / \langle n_k \rangle$ is the compositional distribution within the mean distribution. We begin with the observation that the mean color-blind distribution is the same as in the one-component case. This follows from the fact that the choice of the points at which the string of beads is cut is independent of the compositional makeup of the particle (Figure 1). Thus, $\langle n_k \rangle$ is given by Equation (12) with $M = M_A + M_B$:

$$\frac{\langle n_k \rangle}{N} = \frac{\Omega^{(1)}_{M_A+M_B-k;N-1}}{\Omega^{(1)}_{M_A+M_B;N}} \tag{42}$$

We obtain the compositional distribution by the following construction. Imagine that all possible distributions are stacked vertically to form a table so that column 1 contains the first fragment in all distributions, column 2 contains all second fragments, and so on. All columns are permutations of each other (this follows from the construction of the fragments illustrated in Figure 1) and since all permutations are equally likely (this follows from the condition of random fragmentation), all columns have the same fragment and compositional distribution; therefore, we only need to consider one of them. The equivalent problem now is this: count the number of ways to select $a$ beads from a pool of $M_A$ $A$s and $k - a$ beads from a pool of $M_B$ $B$s and take its ratio over the total number of ways to pick $k$ beads:

$$\overline{c_{a|k}} = \binom{M_A}{a}\binom{M_B}{k-a} \bigg/ \binom{M_A + M_B}{k}. \tag{43}$$

The mean distribution then is the product of the size and compositional distributions:

$$\langle n_{a|k} \rangle = \langle n_k \rangle \, \overline{c_{a|k}}, \tag{44}$$

or

$$\frac{\langle n_{a,b} \rangle}{N} = \frac{\binom{M_A}{a}\binom{M_B}{b}}{\binom{M_A+M_B}{a+b}} \frac{\Omega^{(1)}_{M_A+M_B-a-b;N-1}}{\Omega^{(1)}_{M_A+M_B;N}} \tag{45}$$

It is straightforward algebra to show that this is equivalent to Equation (37).

For $M_A \gg a$, $M_B \gg b$, the compositional distribution goes over to the binomial:

$$\frac{\binom{M_A}{a}\binom{M_B}{b}}{\binom{M_A+M_B}{a+b}} \rightarrow \binom{a+b}{a} \phi_A^a \phi_B^b, \tag{46}$$

with $\phi_A = M_A/(M_A + M_B)$, $\phi_B = 1 - \phi_A$. It is a more compact expression than Equation (43), but is valid only in the asymptotic limit.

Figure 2 shows compositional distributions for a bicomponent particle with $M_A = 4$ units of $A$ and $M_B = 3$ units of $B$. As a means of a demonstration we show the results of a Monte Carlo simulation, which are seen to be in excellent agreement with theory. The binomial distribution, also shown for comparison, is only in qualitative agreement because the fragment masses are small and the conditions for asymptotic behavior are not met in this case.

### 2.3. Any Number of Components

Extension to any number of components follows in a straightforward manner from the bicomponent case but the notation becomes less transparent. Suppose the parent particle consists of $K$ components $A, B \cdots$ and contains $M_A$ units of $A$, $M_B$ units of $B$, and so on. The distribution of fragments is now expressed by the $K$-dimensional vector $\mathbf{n} = \{n_{a,b\ldots}\}$ that gives the number of fragments that contain $a$ units of component $A$, $b$ units of $B$, etc. This distribution satisfies

$$\sum_{a,b\cdots} n_{a,b\cdots} = N \tag{47}$$

$$\sum_{a,b\cdots} z n_{a,b\cdots} = M_Z; \quad z = a, b \cdots \tag{48}$$

where $M_Z$ is the mass of component $z = a, b \cdots$ in the parent particle. The set of all distributions that satisfy the above conditions constitutes the ensemble of all distributions that are produced by the fragmentation of the parent particle into $N$ fragments.

Random fragmentation is once again implemented as shown in Figure 1: Given a string of colored beads, we cut it at $N - 1$ random points to produce $N$ fragments. All permutations of the beads are equally probable. Accordingly, configurations are equally probable. The number of configurations is

$$\Omega_{\mathbf{M};N}^{(K)} = \mathbf{M}! \, \Omega_{M;N}^{(1)} = \left( \frac{M!}{M_A! M_B! \cdots} \right) \binom{M-1}{N-1}, \tag{49}$$

where $M = M_A + M_B + \cdots$ is the total mass of the particle and $\mathbf{M}! = (M_A, M_B \cdots)!$. The multiplicity $\omega(\mathbf{n})$ of distribution $\mathbf{n}$ is the number of configurations with that distribution and is given by the number of permutations in the order of fragments and in the order of components within each fragment:

$$\omega(\mathbf{n}) = \mathbf{n}! \prod_{a,b\cdots} \left( \frac{(a + b + \cdots)!}{a! b! \cdots} \right)^{n_{a,b\cdots}} = \mathbf{n}! \prod_{\mathbf{c}} (\mathbf{c}!)^{n_{\mathbf{c}}} \tag{50}$$

with $\mathbf{c} = (a, b \cdots)$. The probability of distribution $\mathbf{n}$ is

$$P(\mathbf{n}) = \frac{\omega(\mathbf{n})}{\Omega_{\mathbf{M};N}^{(K)}} \tag{51}$$

and the partition function is the sum of multiplicities of in the ensemble:

$$\Omega_{\mathbf{M};N}^{(K)} = \sum_{\mathbf{n}} \omega(\mathbf{n}). \tag{52}$$

The mean distribution of fragments is

$$\frac{\langle n_{\mathbf{c}|k} \rangle}{N} = \mathbf{c}! \frac{\Omega_{\mathbf{M}-\mathbf{c};N-1}^{(K)}}{\Omega_{\mathbf{M};N}^{(K)}} \tag{53}$$

and is the generalization of (37). Alternatively, the mean distribution can be expressed by analogy to Equation (44) as the product of the color blind distribution with a mean compositional distribution:

$$\langle n_{\mathbf{c}|k} \rangle = \langle n_k \rangle \, \overline{c_{\mathbf{c}|k}}. \tag{54}$$

The mean color-blind size distribution $\langle n_k \rangle / N$ is the same as in one-component fragmentation,

$$\frac{\langle n_k \rangle}{N} = \frac{\Omega_{M-k;N-1}^{(1)}}{\Omega_{M;N}^{(1)}} \tag{55}$$

with $M = M_A + M_B + \cdots$, $k = a + b + \cdots$, and $\overline{c_{\mathbf{c}}}$ is the conditional probability that the compositional vector of fragment size $k$ in the mean distribution is $\mathbf{c} = (a, b \cdots)$:

$$\overline{c_{\mathbf{c}|k}} = \binom{M_A}{a} \binom{M_B}{b} \cdots \bigg/ \binom{M}{a + b + \cdots}. \tag{56}$$

This is the generalization of Equation (43).

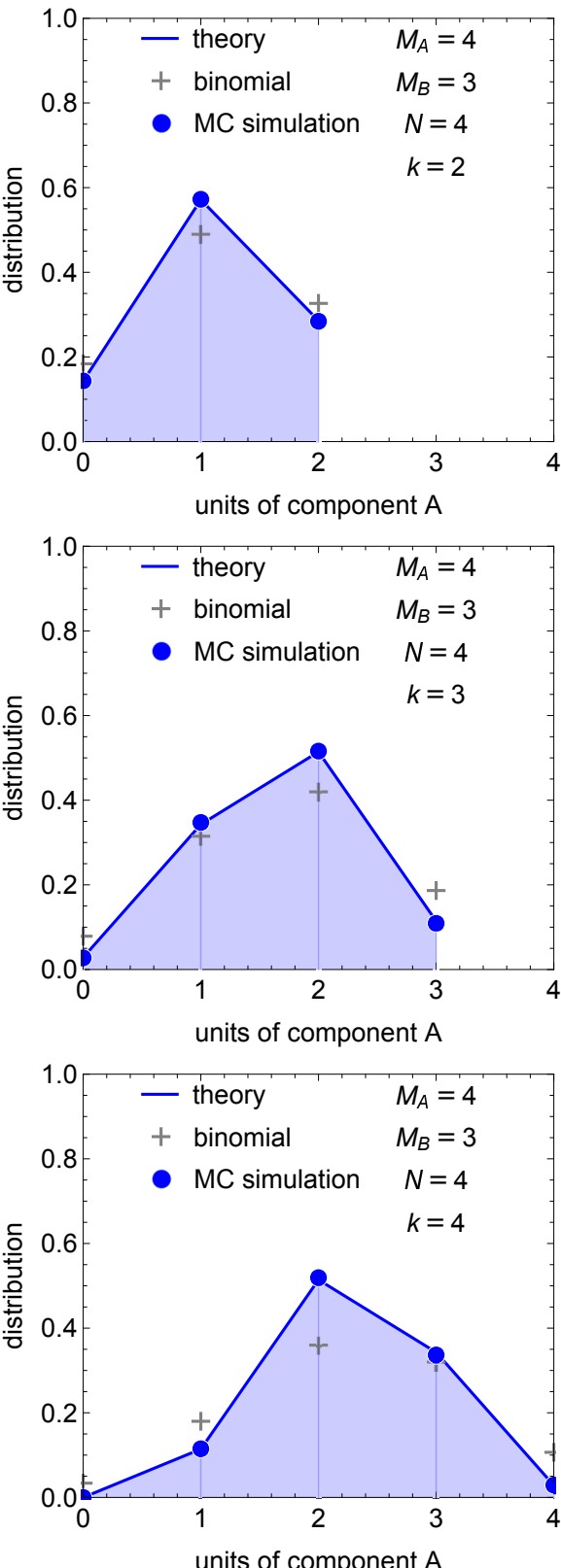

**Figure 2.** The compositional distribution $\overline{c_{a|k}}$ in particles of mass $k = 2, 3,$ and $4$. The parent particle contains $M_A = 4$ units of $A$, $M_B = 3$ units of $B$, and breaks into $N = 4$ pieces. Lines are from Equation (37) and points are from MC simulation after 20,000 fragmentation events. The excellent agreement between MC and theory demonstrates the exact nature of Equation (37) and validates the MC method. The binomial distribution is only asymptotically valid and in this case is a poor approximation because the size of the fragments is small.

### 3. Nonrandom Bicomponent Fragmentation

In random fragmentation, all permutations are equally probable. We now bias the probability of the permutation by a functional $W(\mathbf{n})$ of the fragment distribution such that the probability of fragment distribution $\mathbf{n}$ is

$$P(\mathbf{n}) = \frac{\omega(\mathbf{n})W(\mathbf{n})}{\tilde{\Omega}^{(K)}_{M_A,M_B;N}} \tag{57}$$

with

$$\tilde{\Omega}^{(K)}_{M_A,M_B;N} = \sum_{\mathbf{n}} \omega(\mathbf{n})W(\mathbf{n}). \tag{58}$$

and $\omega(\mathbf{n})$ from Equation (50). Here, $\omega(\mathbf{n})$ is the intrinsic multiplicity of $\mathbf{n}$ in the ensemble of fragments, while the product $\omega(\mathbf{n})W(\mathbf{n}) \doteq \tilde{\omega}(\mathbf{n})$ is its apparent (biased) multiplicity as weighted by the bias functional and distinguished by the tilde. Similarly, the partition function $\tilde{\Omega}$ is the summation of the apparent (biased) multiplicities of all distributions in $\mathscr{E}_{M_A,M_B;N}$. All permutations in the same configuration of fragments are equally probable under this formulation, as they all have the same distribution $\mathbf{n}$. With $W = 1$, we recover the random case (all permutations in all configurations are equally probable). Accordingly, "random" and "unbiased" both refer to uniform bias $W = 1$.

#### 3.1. Linear Ensemble

The bias functional $W$ will remain unspecified. This allows us to choose the bias so as to produce any desired distribution of fragments. A special but important case is when $W$ is of the product form

$$W(\mathbf{n}) = \prod_a \prod_b (w_{a,b})^{n_{a,b}}, \tag{59}$$

where $w_{a,b}$ are factors that depend on $a$ and $b$ but not on the fragment distribution itself. The log of the bias is then a linear functional of $\mathbf{n}$:

$$\log W(\mathbf{n}) = \sum_a \sum_b n_{a,b} \log w_{a,b}. \tag{60}$$

The result states that the log of the bias is homogeneous functional of $\mathbf{n}$ with degree 1, i.e., $\log(\lambda \mathbf{n}) = \lambda \log W(\mathbf{n})$ for any $\lambda > 0$. We refer to this case as linear bias with the understanding that linearity actually refers to the log of $W$.

**Proposition 5.** *The mean distribution of fragments under the bias in Equation (59) is*

$$\frac{\langle n_{a,b} \rangle}{N} = w_{a,b} \binom{a+b}{a} \frac{\tilde{\Omega}^{(2)}_{M_A-a,M_B-b;N-1}}{\tilde{\Omega}^{(2)}_{M_A,M_B;N}} \tag{61}$$

*with*

$$\tilde{\Omega}^{(2)}_{M_A,M_B;N} = N! \sum_{\mathbf{n}} \prod_{a=0}^{\infty} \prod_{b=0}^{\infty} \frac{w_{a,b}^{n_{a,b}}}{n_{a,b}!} \binom{a+b}{a}^{n_{a,b}} \tag{62}$$

*and the summation over all $\mathbf{n}$ that satisfy Equations (18)–(20).*

**Proof.** We write the apparent multiplicity $\tilde{\omega}(\mathbf{n})$ of distribution $\mathbf{n}$ as

$$\tilde{\omega}(\mathbf{n}) = N! \prod_{a=0}^{\infty} \prod_{a=b}^{\infty} \frac{(\alpha_{a,b})^{n_{a,b}}}{n_{a,b}!}, \tag{63}$$

and the probability of distribution as

$$P(\mathbf{n}) = \frac{N!}{\tilde{\Omega}_{M_A,M_B;N}} \prod_{a=0}^{\infty} \prod_{a=b}^{\infty} \frac{(\alpha_{a,b})^{n_{a,b}}}{n_{a,b}!}, \tag{64}$$

with

$$\alpha_{a,b} = w_{a,b} \binom{a+b}{a}, \tag{65}$$

The claim of Proposition 5 then follows directly from Proposition 4. □

### 3.2. Composition-Independent Bias

If the bias factors are of the form $w_{a,b} = g_{a+b}$, where $g_k$ is a function of a single variable, the acceptance probability of a configuration of fragments depends on the mass $k = a + b$ of the fragment but not on its composition. This leads to a simple expression for the mean distribution by the following argument. With reference to Figure 1, fix the points where the string is cut; this amounts to fixing the color blind distribution of the fragments. All permutations of components are equally probable because they have the same distribution. Accordingly, the compositional distribution is the same as in the random case and is given by Equation (43). The size distribution on the other hand is biased and is the same as when the same bias is applied to one-component distribution. The final result is

$$\frac{\langle n_{a,b} \rangle}{N} = \frac{\binom{M_A}{a}\binom{M_B}{b}}{\binom{M_A+M_B}{a+b}} \frac{\langle n_{a+b} \rangle}{N}, \tag{66}$$

where $\langle n_{a+b} \rangle = \langle n_k \rangle$ is the one-component size distribution under bias $w_{a,b} = g_{a+b}$,

$$\frac{\langle n_k \rangle}{N} = g_k \frac{\tilde{\Omega}^{(1)}_{M_A+M_B-k;N-1}}{\tilde{\Omega}^{(1)}_{M_A+M_B;N}} \tag{67}$$

with

$$\tilde{\Omega}^{(1)}_{M_A+M_B;N} = N! \sum_{\mathbf{n}} \prod_{k=1}^{\infty} \frac{g_k^{n_k}}{n_k!}. \tag{68}$$

Except for special forms of $g_k$ the partition function will not be generally available in closed form. Table 1 summarizes three cases for which exact results are possible. All three cases are associated with distributions encountered in binary aggregation [35]. The partition functions in cases 1 and 2 refer to the constant and sum kernels, respectively, and are exact; case 3 is associated with the product kernel and gives the asymptotic limit of the partition function for $M, N \gg 1$, $M/N < 2$, conditions that refer to the pre-gel state [36].

In the general case, $w_{a,b}$ depends on both $a$ and $b$ explicitly and affects both the size and compositional distributions. This case will be demonstrated by simulation in the next section.

**Table 1.** Closed form results for three composition-independent bias functionals.

|  | $w_{a,k-a}$ | $\Omega^{(1)}_{M;N}$ |
|---|---|---|
| Case 1 | $1$ | $\binom{M-1}{N-1}$ |
| Case 2 | $\frac{k^{k-1}}{k!}$ | $m^{M-N}\frac{N!}{M!}\binom{M-1}{N-1}$ |
| Case 3 ‡ | $2\frac{(2k)^{k-2}}{k!}$ | $\left(m^{M-N}\frac{N!}{M!}\right)^2\binom{M-1}{N-1}$ |

‡ asymptotically for $M, N \gg 1$, $M/N < 2$.

## 4. Simulation of Biased Fragmentation

Except for certain special forms of the bias the mean fragment distribution cannot be calculated analytically and the only recourse is stochastic simulation. Here, we describe a Monte Carlo (MC) algorithm for sampling the ensemble of distributions. We will then use this method to demonstrate results for two cases of biased bicomponent fragmentation.

### 4.1. Monte Carlo Sampling by Exchange Reaction

Suppose $\mathbf{m} = ((a_1, b_1), \cdots (a_N, b_N))$ is a configuration of $N$ bicomponent fragments, such that fragment $i$ contains $a_i$ units of component $A$ and $b_i$ units of component $B$. The probability of configuration is equal to the probability of its distribution, $P(\mathbf{n})$, divided by its multiplicity $\omega(\mathbf{n})$; from Equation (57) this probability is

$$\text{Prob}(\mathbf{m}) = \frac{W(\mathbf{m})}{\Omega^{(2)}_{M_A, M_B; N}}, \tag{69}$$

where $W(\mathbf{m}) = W(\mathbf{n})$ is the bias of configuration $\mathbf{m}$, equal to the bias of its distribution $\mathbf{n}$. If the bias functional is of form in Equation (59), its value on configuration $\mathbf{m}$ is

$$W(\mathbf{m}) = \prod_{i=1}^{N} w_{a_i, b_i}. \tag{70}$$

Here, the product is over the $N$ fragments in the configuration, whereas in Equation (59) it is over all all units $a$ and $b$ in the distribution. Suppose that two fragments $i$ and $j$ exchange mass according to the reaction

$$(a_i, b_i) + (a_j, b_j) \to (a_i', b_i') + (a_j', b_j') \tag{71}$$

under the conservation conditions $a_i + a_j = a_i' + a_j'$ and $b_i + b_j = b_i' + b_j'$. This amounts to a transition $\mathbf{m} \to \mathbf{m}'$ between configurations with equilibrium constant

$$\mathcal{K}_{\mathbf{m} \to \mathbf{m}'} = \frac{\text{Prob}(\mathbf{m}')}{\text{Prob}(\mathbf{m})} = \frac{W(\mathbf{m}')}{W(\mathbf{m})} = \frac{w_{a_i', b_i'} w_{a_j', b_j'}}{w_{a_i, b_i} w_{a_j, b_j}}. \tag{72}$$

The stationary distribution of this exchange reaction is the same as that in Equation (57) [35]. Accordingly, the ensemble of fragment distributions may be sampled via exchange reactions by tuning the equilibrium constant to the selection functional according to Equation (72).

To implement this sampling computationally, we represent fragments as a list of 1s (representing component $A$) and 0s (component $B$). We pick two clusters $i$ and $j$ at random, merge them into a single list, randomize the order of components, and break them into two new fragments by picking a break point at random. We accept the resulting configuration by the Metropolis criterion based on the equilibrium constant in Equation (72): we accept the result of the exchange if rnd $\leq \mathcal{K}_{\mathbf{m} \to \mathbf{m}'}$, where rnd is a random number uniformly distributed in $(0, 1)$; otherwise we reject. With $W = 1$, every exchange reaction is accepted, which amounts to random fragmentation. The randomization of the order of

components in the merged list ensures that all permutations are considered with equal probability. If $W = 1$, the resulting configuration is always accepted and the distribution conforms to random fragmentation. This is how the MC results in Figure 2 were obtained and the agreement with the theoretical distribution serves as a validation of the numerical algorithm.

*4.2. Two Examples*

In random fragmentation ($w_{a,b} = 1$), the compositional distribution is given by Equation (43). We may choose the bias functional so as to produce deviations in either direction relative to the random case. It is possible to produce positive deviations (preferential segregation of components in the fragments relative to random mixing) or negative deviations (more intimate mixing than in random mixing). We demonstrate both behaviors using the two examples below:

1.  Case I (positive deviations)

$$w_{a,b} = (a+1)^\alpha + (b+1)^\alpha \tag{73}$$

2.  Case II (negative deviations)

$$w_{a,b} = (a+1)^\alpha (b+1)^\alpha \tag{74}$$

In case I, the fragment bias $w_{a,b}$ is an additive function of the amounts of the two components. Considering that $a + b$ is constrained by mass balance, the fragment bias is large for fragments that are rich in either component but small for fragments that are relatively mixed. This ought to favor the formation of fragments in which the components are relative segregated. The fragment bias in case II is a multiplicative function of the amounts of the two components. It is large in fragments that contain both components but quite small if one component is present in excess of the other. This form ought to produce fragments that are better mixed than fragments produced by random fragmentation.

We test these behaviors in Figure 3 which shows results for $\alpha = 4$. In this example the particle contains an equal number of units of each component, $M_A = M_B = 20$, and breaks into $N = 4$ pieces. In both cases the size distribution deviates from that in random fragmentation. Compositional distributions are shown for sieve-cut masses $k = 2$, 4, and 8. The additive bias (case I) produces distributions that are more spread out relative to the random case. For $k = 2$, in particular, the compositional distribution is inverted relative to the random case. This indicates strong segregation, as the majority of fragments contains either $A$ or $B$, while only few fragments in this size contain both components. As the fragment size increases the segregation of components is less pronounced, though always present, as indicated by the fact that the random distribution is always narrower. The opposite behavior is observed in case II: distributions are narrower than those in random fragmentation, especially in the smaller fragment sizes. In this case, the fragments are better mixed relative to the parent particle. As a general trend in both cases, deviations from random mixing are most pronounced in small fragment sizes. Large fragments on the other hand are close to randomly mixed. There is simply not enough material to produce a large fragment that is highly enriched in one component; thus mixing prevails. This limitation is not present in small clusters.

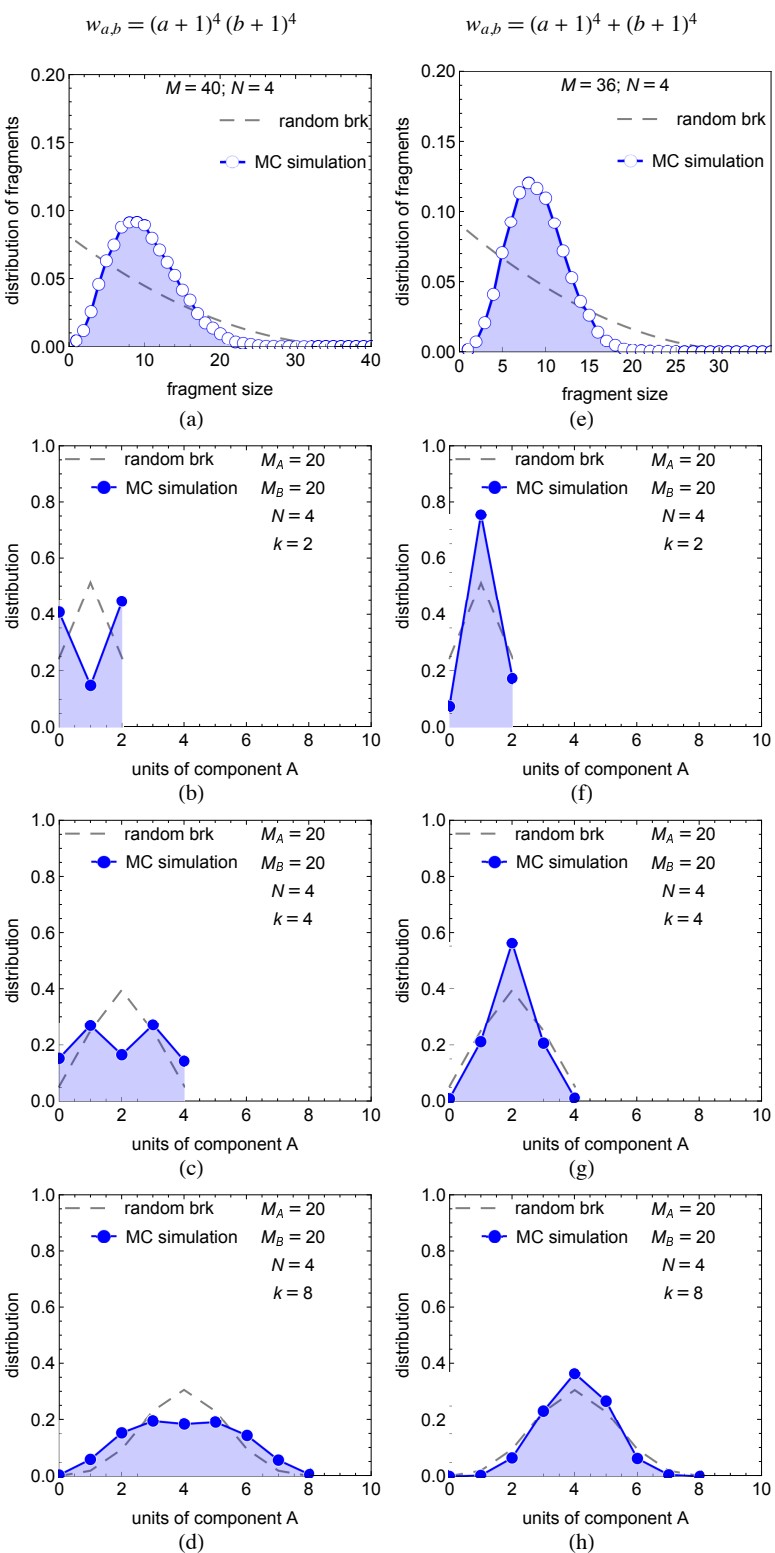

**Figure 3.** Size and compositional distributions at fragment sizes $k = 2, 4$, and 8 for two bias functionals: (**a–d**): $w_{a,b} = (1 + a)^4(1 + b)^4$; (**e–h**): $w_{a,b} = (1 + a)^4 + (1 + b)^4$. In both cases, the particle contains $M_A = 20$ units of $A$, $M_B = 20$ units of $B$, and breaks into $N = 4$ fragments.

## 5. Concluding Remarks

We have presented a treatment of multicomponent fragmentation on the basis of random fragmentation in combination with a functional that biases the ensemble of feasible distributions.

The two key notions in this treatment are the set of feasible distributions and the multiplicity of distribution within this set as established by the rules that define "random" fragmentation. In the random-fragmentation ensemble, distributions are proportional to their multiplicity. This problem is analytically tractable and we have presented its solution for any number of components and fragments. A third key notion is that of the bias functional that modulates the probability of feasible distributions and allows us to obtain fragment distributions other than that of random fragmentation with deviations in either direction.

Random fragmentation is not endowed with any special universality. In certain problems, such as the linear chain in Figure 1, selecting the bonds to break at random might be a reasonable physical model and random fragmentation applies; in general though, this will not be the case. The importance of random fragmentation is mathematical. Similar to the "fair coin" or the "ideal solution," it provides an analytically solvable reference case from which to measure deviations in systems that do not conform to this model. The tool that quantifies these deviations is the bias functional. This functional, analogous to the activity coefficient in solution thermodynamics [37], permits the systematic construction of distributions that exhibit any degree of deviation from the random case. This is a key result of this formulation.

In single-component fragmentation, the quantity of interest is the mean size distribution of the fragments. In multicomponent systems we are additionally concerned with the compositional distribution of the fragments. This introduces a new dimension to the problem and raises questions of mixing and unmixing of components. Do fragments inherit the compositional characteristics of the parent particle? Do they become progressively more well mixed or less? Both behaviors are possible and are quantified via the bias functional $W$. This functional is where the mathematical theory of fragmentation presented here makes contact with the physical mechanisms that lead to the disintegration of material particles. To make this connection quantitatively, one must begin with the a physical model of fragmentation that assigns probabilities to all possible distributions of fragments that can be generated. This is a major undertaking and is specific to the particular problem that is being considered. The point we wish to make is that once such results are available, their reduction into a compositional distribution passes through the bias functional, which represents the contact point between physics and the mathematical formulation of fragmentation.

Lastly, the connection to statistical mechanics should not be lost. We have constructed an ensemble whose fundamental element ("microstate") is a the ordered configuration of fragments; its total number in the ensemble is the partition function. The higher-level stochastic variable (the observable) is the distribution of fragments and its probability is determined by its multiplicity in the ensemble. The form of the probability in Equation (13), also known as Gibbs distribution [27], is encountered in time reversible processes as well as in population balances of aggregation and breakup [24–27,38]. The derivation of the mean distribution in the random case follows in the steps of the Darwin–Fowler method [39]. Additionally, the compositional distribution in random breakup is given asymptotically by the binomial distribution in Equation (46). This establishes a reference for compositional interactions analogous to that of the ideal solution in thermodynamics. In fact, the Shannon entropy of the binomial distribution is the ideal entropy of mixing when two pure components coalesce into a single particle that contains mass fraction $\phi_A$ of component $A$. These connections are not coincidental. Biased sampling from a distribution generates a probability space of distributions and when the base distribution is exponential, this ensemble obeys thermodynamics [40]. In fragmentation the base distribution is a multicomponent exponential: the size distribution in Equation (12) goes over to the exponential distribution when $M, N \gg 1$. In this limit, the ensemble of fragments becomes mathematically equivalent to a thermodynamic ensemble of two components with interactions that produce positive or negative deviations relative to ideal solution.

**Funding:** This research received no external funding.

**Conflicts of Interest:** The author declares no conflict of interest.

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
