# Peer review of "Statistical Mechanics of Discrete Multicomponent Fragmentation"

_condensedmatter, doi:10.3390/condmat5040064_

Round 1

Reviewer 1 Report

This is a very interesting and sophisticated paper that makes a connection between ensembles of randomly fragmented particles and biased fragmentation.  The theory is clearly explained, and although a bit involved, it is understandable to a general reader.  The author confirms his predictions for the biased cases by a well-thought-out computer simulation.  I recommendation.

Here are a few questions/comments:

1.  Fig. 1: You can add that in this example, N = 3 and there are (9 choose 2) =36 ways to partition this system.

2. How is P(n) of equation (4) normalized?  Sum over all sets n  equals 1?  (Perhaps say explicitly)

3. Above equation (15): the derivative with respect to alpha_k I believe.  Also, shouldn’t that be the partial derivative?

4. Equation (15) — shouldn’t alpha_k be in the denominator since we are differentiating alpha_k^(n_k)?  (It doesn’t affect the results of course because we set alpha_k=1.

5. Equation (16) — also partial derivative. Should that be alpha_k in the denominator?

6. Table 1:  Should the bias functions omega_{a,k-a} be the same for cases 2 and 3?  (I think one). 

Reviewer 2 Report

This author analyzes the mass or size distribution of various fragmentation processes in terms of equiprobable (or biased) configurations, from a statistical point of view, and provides exact formulas.

Equation (17) was derived in the literature as the author mentioned, but the interesting parts is the probability for multiple components, sections 2.2 and 2.3, as well as the biased fragmentation studied in section 3. The author does not include the dynamics here, since the fragmentation stops after one step, and averages are performed on the different possible configurations obtained. Many papers treat the dynamical case, where the fragmentation continues by breaking the smaller elements and where averages are done after a long time an rescaling. However it is an interesting paper and certainly deserves publication. Just few remarks:

1/ I would write "m" (or "n") instead of "N" fragments, and "M" instead of "m" for clarity because N<=m in the text and it would be more easier to read with for example "m<=M", but this is only my point of view.

2/ Fig 1: K=3 should be N=3

3/ P(n) and P(m) can be confusing in equation (8), maybe write Prob(.) instead

4/ In figures 3(a) and 3(b), it would be more interesting to draw the log or log-log plots, to see the forms of the asymptotes.

Reviewer 3 Report

report on “Statistical Mechanics of Discrete Multicomponent Fragmentation” by Themis Matsoukas

The Paper is of interest, but many errors have to be corrected and the text needs to be completed.

. Figure 1 remove “K=3”

. line 84: sentence with no verb

. Eq (12) numerator is not N-1 but N-2

. which means that the proof is wrong, at least because the case n_k=0 is not treated properly

. sentence over Eq (15): the derivative is with respect to alpha_k not n_k

. Eq (15) the factorial sign at the denominator should not be superscript

. Eq (15) alpha_k should be at the denominator at the center and the right-hand side

. Eq (17) <n_k>, not n_k

. line 101 k = a+b, not k = a_b

. for a better readability, insert a small space $\,$ between the variables, in all equations

. Eq (22) and Eq (23) the index is k, not i

. Eq (24) the k max is m

. the equations after Eq (24): the sums over a and k must be swapped

. line 109 “Random fragmentation is”, not “in”

. Eq (31) is the same as Eq (27), it must be removed

. Eq (34): the product on k starts at 1, not 0

. Eq (41): the (2) superscripts are missing

. line 122 “Here is”, not “Here in”

. Eq (42) replace -a-b by -k

. Eq (43) <c_a|k> is a misleading notation since, unlike Eq (11) for example, it is not sum_n c_a|k(n) P(n) because the terms with n_k=0 have to be removed from the sum. This also changes the normalization. As defined in the paper, this average is: sum_{n/n_k(n) /= 0} c_a|k(n) P(n) / sum_{n/n_k(n) /= 0} P(n)

. Eq (46) replace k by a+b

. line 129 “The set of all”

. line 131 “into N fragments”

. Eq (51) the arrow is in fact an omega

. line 132, 133: already said

. Eq (54) n_{c|k} not n_c

. 135 “Nonrandom fragmentation with two colors”

. Eq (60) swap n_{a,b} before the log

. Eq (62) and Eq (64) replace k by a+b

. Eq (65) replace k by a+b and k-a by b

. Section 4.1, the Metropolis algorithm does not take into account the proposal distributions p(x’|x) and p(x|x’) corresponding to transition (70) and its reverse transition. This has to be justified.

A lot of commas are missing (for example, none in the abstract) which make the text less fluid to read. The author should choose between “components” and “colors”.

This paper concerns the so-called “1D bond percolation” so this expression should appear in the text. The introduction seems to ignore the literature about the works on fragmentation that are neither 1D, nor single component, nor binary, that have been produced in the fields of nuclear physics, cluster physics, percolation and molecular physics such as, for example:

Nuclear physics: Anthony John Cole (1995, 2004), Dieter Gross (1995, 1997), O Shapiro (1997)

Cluster physics: Paul-Antoine Hervieux (2001), S. Diaz-Tendero (2005, 2006)

Percolation: the literature is very rich, for example, the same work at infinite dimension: P Desesquelles (2011), Antoine Alard (2012,2015)

Molecules: Denis Tikhonov (2019), P Desesquelles (2020) and many others, this is a hot subject presently.

Reviewer 4 Report

In this manuscript, the author gives a combinatoric approach to understand
the fragment size statistics is an idealized and discrete fragmentation
model. Overall, the manuscript is well written and well produced, so I have
little to critique from the literary perspective.

However, I do have some comments for the author's consideration:

1. Is there some physical motivation for considering a bicomponent (or a
multicomponent) system? I appreciate that multicomponents represent a
natural mathematical extension of the basic system, but a physical motivation
would be useful.

2. The distribution on the left-hand side of Eq. (43) has not been clearly
defined. Please do so. Because of the missing definition, I'm unable to
understand the meaning of Fig. 2. I don't understand the meaning of the
vertical axis and the data points marked "binomial" in the figure. The
results look simple, so the figure should be simple to understand. Also,
what wisdom is one supposed to learn from the figure? All I can glean is
that simulations and theory agree, but I don't know what controls the shape
of the distribution from a physical perspective.

The remainder of the paper treats more components and then the role of a bias
functional on the fragment size distributions. All this is worked out
competently, but as an outsider to this problem, I have no idea why a reader
should care about these results. It would be beneficial for the author to
attempt to address this concern so that there might be some appeal to a wider
readership than merely expert in fragmentation theory.

Round 2

Reviewer 1 Report

The author has addressed my concerns and suggestions, as well as those of the other authors, and I recommend publication.

Author Response

The author has addressed my concerns and suggestions, as well as those of the other authors, and I recommend publication.

I thank the reviewer

Reviewer 3 Report

Eq (16): there is a problem with writing (nk-1)! since nk can be zero in some distributions n. However, in this case, the derivative of alphak^nk would be zero, so these terms vanish. Nevertheless, it would be more rigorous to write the sum as : sum_{n | nk /= 0}

Eq(54): I still don’t agree with the author. The equation gives the average compositional distribution for the fragments with k elements, thus k must appear as an index of n_{c|k} in the same way as in Eq (44) the left hand side is <n_{a|k}>, not <n_a>

Author Response

Eq (16): there is a problem with writing (nk-1)! since nk can be zero in some distributions n. However, in this case, the derivative of alphak^nk would be zero, so these terms vanish. Nevertheless, it would be more rigorous to write the sum as : sum_{n | nk /= 0}

Author's Response: As the Reviewer says, if n_k=0 the terms not contribute tot he summation. I agree that for completeness the condition n_k\neq 0 must be added and I have now included it under the summation in Eq. 16.

Eq(54): I still don’t agree with the author. The equation gives the average compositional distribution for the fragments with k elements, thus k must appear as an index of n_{c|k} in the same way as in Eq (44) the left hand side is <n_{a|k}>, not <n_a>

Author's Response: The reviewer is correct. I have added the notation "|k" in Eqs 53 and 54.

Again I thank thank the reviewer for reading the manuscript so carefully.